# Being a Dog: A Review of the Domestication Process

**DOI:** 10.3390/genes14050992

**Published:** 2023-04-27

**Authors:** Domenico Tancredi, Irene Cardinali

**Affiliations:** Department of Chemistry, Biology and Biotechnology, Università degli Studi di Perugia, 06123 Perugia, Italy; domenicotancredi94@gmail.com

**Keywords:** dog, *Canis familiaris*, domestication, wolves, mitochondrial DNA, Y chromosome, Mediterranean area

## Abstract

The process of canine domestication represents certainly one of the most interesting questions that evolutionary biology aims to address. A “multiphase” view of this process is now accepted, with a first phase during which different groups of wolves were attracted by the anthropogenic niche and a second phase characterized by the gradual establishment of mutual relationships between wolves and humans. Here, we provide a review of dog (*Canis familiaris*) domestication, highlighting the ecological differences between dogs and wolves, analyzing the molecular mechanisms which seem to have influenced the affiliative behaviors first observed in Belyaev’s foxes, and describing the genetics of ancient European dogs. Then, we focus on three Mediterranean peninsulas (Balkan, Iberian and Italian), which together represent the main geographic area for studying canine domestication dynamics, as it has shaped the current genetic variability of dog populations, and where a well-defined European genetic structure was pinpointed through the analysis of uniparental genetic markers and their phylogeny.

## 1. Introduction

Domestication represents the evolutionary process in which the genetic, physiological, and behavioral (and cognitive) profile of a species is reshaped to adapt to a man-made environment. It arises from mutualism between two species, where the domesticator creates an environment and actively manages both the survival and reproduction of another species (the domesticated), which provides the former with resources and/or services [1].

The selective pressures involved in domestication are likely to have varied over time and across taxa, although it is believed that in many species, the early stages of this process were largely dependent on the evolutionary reduction of fear and aggression toward humans [2]. The domestication of plants and animals represents undoubtedly one of the greatest achievements mankind has ever made, as the cultivation of “selected” plant species enabled humans to increase their food supply, while the domestication of animal species made it easier to perform diversified tasks, such as hunting and managing cultivated land. The domestication of surrounding flora and fauna by ancient human populations required an intentionality and an understanding of other species’ behavior and reproductive biology [3,4]; thus, the cognitive demands of the human-mediated domestication process constitute a phenomenon distinct from other interspecific mutualisms, i.e., those evolved by social insects [5,6].

The massive domestication of plants and animals by humans was likely triggered by the significant climatic and environmental changes that characterized the global transition from the Last Glacial Maximum (LGM) to the current Holocene interglacial period [7]. The demographic pressure produced by the exponential growth of the human population was proposed as a random factor favoring domestication, since it produced an increase in relationships between humans and animals.

In some species, such as dogs (*C. f.*), domestication fostered not only social tolerance toward humans, but also active social bonds and relationships rooted in attachment [8]. A variety of both paleontological and genetic evidence has been collected attesting to the dog being one of the first (or the first ever) animals to be domesticated thousands of years before the human transition from hunter-gatherer to an agriculture-based lifestyle. Other domesticated animal species appeared later, about 10,000 years ago, when the dog was already a domesticated and integrated part of human social life [9]. Among all the animal species domesticated by humans, dogs show the most pronounced morphological and behavioral variability currently observable. They are able to perform different tasks within the human social sphere, such as acting as companion animals and assisting disabled people or becoming faithful supporters of the police force.

During the past three decades, studies of population and forensic genetics, ancient DNA (aDNA), and more recently, paleogenomics have provided abundant evidence in support of Charles Darwin’s thesis that domesticated animal populations and the process of domestication of wild populations represent perfect models for understanding evolutionary processes on a large temporal scale [10].

## 2. Canine Domestication Process

The first domesticated taxa were dogs, which diverged from their main ancestor, the gray wolf (*Canis lupus*), between 32,000 and 11,000–16,000 years ago [11,12]. This phenomenon was probably characterized by two functionally distinct phases. The first involved the opening of an anthropogenic niche, which facilitated new and lasting associations between some wolves and human populations. Some theories suggest that wolf populations were first attracted to human-generated waste, which was considered a new food ecology exploited by those wolves that did not show fear of human communities [13]. Other theories propose that human–wolf interactions began with the capture and breeding of newborn wolves carried out by humans [14,15] or through mutualism resulting from the complementary hunting strategies of Paleolithic humans and wolves [16]. Regardless of the circumstances that fostered such human–wolf associations, it can be said that wolf populations initially occupied these ecological niches as “synanthropes,” i.e., non-domesticated species that benefit from an anthropized environment. The second phase of domestication was characterized by a gradual change in “human–dog” interactions, and dogs began to be selected for behavioral characteristics. During this phase, it is likely that social bonds between individual dogs and humans became increasingly important, perhaps favoring those specimens that were biologically better prepared to develop such interspecific relationships [13]. Although wolves can develop relationships with humans if fed “by hand” from puppyhood [15,17], domestication appears to have relaxed the conditions required for such relationships in dogs [13]. While the early stages of the domestication process likely aimed to reduce the fear and the anxiety required for living in an anthropogenic environment, in later stages the selection may have acted more specifically on socioemotional processes related to interspecific social bonding and cooperation [13].

### 2.1. Sources of Selective Pressure: The Environment and Humans

Regardless of the initial process, domestication took place for about 15,000–30,000 years [18,19]. Dogs firstly associated with hunter-gatherers, then humans organized themselves into small settlements, and finally, into larger villages. The analysis of the domestication process, which produced marked differences between wolves and dogs, individuated three types of selective pressures (natural and sexual, human-mediated and improved selection), as depicted by Range and Marshall-Pescini in 2022 (Figure 1) [20].

At the beginning, the natural and sexual selection shaped, and continues to shape, the cognitive and behavioral spheres of dogs. Many studies highlight genetic adaptations to specific environments [19,21] and preferences in the mating of wild dogs [22]. Then, humans “negatively” selected some specific characteristics (consciously or not). This selection allowed coexistence between dogs and humans for many thousands of years (for example, overly aggressive animals were killed). In the end, humans deliberately and directly “positively” selected specific phenotypic traits, a process that began with selecting dogs for different functions, became popular during the Victorian era, and nowadays, is reflected in the multitudes of dog breeds [20].

However, it should be noted that alongside the traits selected by man, the adaptation of dogs to a new ecological niche represents a fundamental part of the domestication process, since cognitive abilities are shaped by the social ecology of species, which differs greatly in the case of dogs and wolves [23].

### 2.2. One Experimental Domestication Model: Belyaev’s Foxes

In 1959, the Russian geneticist Dmitri Belyaev began a study to simulate the process of animal domestication. He aimed to create a selective regime that would reflect the conditions envisioned as “critical” for animal training and to understand the biological basis of the phenotypic changes produced by the domestication process, thus proposing that domestication was based essentially on “selection by tamability” [2]. The experiment was designed to record the effects of additionally intense selection for tameness and the correlated emergence of unexpected morphological and genetic traits [24]. To test this hypothesis, Belyaev used silver foxes, a melanistic variant of the red fox (*Vulpes vulpes*), from a Canadian population of farmed foxes selected over more than 50 generations for increased docility [25]. Then, he systematically bred and tested hundreds of these individuals to verify their “friendliness” toward humans [2,26]. Over a span of 60 years, Belyaev and collaborators documented phenotypic changes in “their” foxes that closely resembled those produced by canine domestication [27] and discovered a drastic reduction in blood cortisol concentrations and less adrenal cortex reactivity in foxes selected for tameness compared to wild-type controls [2,28,29] (Figure 2).

As evidenced by Belyaev, domestic animals differ from their “wild” ancestors from the physiological, morphological, and behavioral points of view, and they appear less fearful than humans and much more socially tolerant of conspecifics. All these changes are considered to be deeply connected to the so-called “Domestication Syndrome” [2,27,28,30], a set of different traits associated with early domestication and observed in some domesticated mammals [31]. However, none of these traits can be found in all domestic mammals, which underwent selective pressures that led to a differentiation within species/breed [31,32]. In 2014, Wilkins and colleagues suggested that all variations observed during domestication are the result of a mild neural crest stem cell deficiency during embryonic development (the Neural Crest Cell Hypothesis; NCCH) [33]. In this complex scenario, there has recently been a debate about the recognition of Domestication Syndrome as a model to describe the phenomenon of domestication [25,31,34,35,36,37,38,39,40]. Lord and colleagues argued that focusing only on behavioral selection represents a severe limitation and so invited the scientific community to consider animal adaptations to a human-modified environment in order to adopt a more thorough approach [25]. Other authors stated that even if we assume that Domestication Syndrome exists and responds to a behavioral selection, it could be due to a selection caused by pleiotropy and mediated by trait-specific mechanisms involving (or not) the neural crest [39]. Despite this, the possibility that dogs have become “less selectively social” is supported by recent studies linking “hypersociality” in dogs to genetic variants associated with “Williams–Beuren Syndrome” [41], a neurodevelopmental disorder characterized by extreme gregariousness and attraction to strangers, in which both oxytocin and vasopressin appear to be unregulated [42].

### 2.3. A Possible Role for Oxytocin in Domestication

Oxytocin is a neuropeptide synthesized by the hypothalamus of mammals and all vertebrates [43,44], and it plays a key role in the development of selective social bonds, such as between monogamous dyads or between adults and infants [45,46]. It was observed that oxytocin seems to trigger an increase in social trust [47,48], modulate social attention and different aspects of social engagement, such as eye contact [49,50], and promote behavioral synchrony, which is critical for coordinating group actions [51,52]. Studies performed on primates and rodents suggest that oxytocin also affects serotonin regulation, stimulating its release in limbic regions and increasing the availability of serotonin receptors [53,54,55].

Based on data combining the variation in the serotonin concentration in the blood of Belyaev’s foxes [56] with the well-known role of serotonin in inhibiting aggressive behavior [57,58], it was inferred that oxytocin–serotonin interactions also contribute to the reduction of aggressive behavior [13]. Herbeck and colleagues hypothesized that in the early stages of domestication, the most important functions covered by oxytocin were those associated with an attenuation of stress reactivity and an inhibition of aggressive behavior. However, the effects of this neuropeptide are known to be highly context-dependent and moderated by a wide range of biological and social factors, and such phenomena have been recently observed in domestic dogs [13].

During calm and affiliative human–animal interactions, both humans and dogs show increased blood concentrations of oxytocin [59,60,61,62]. In the contexts of interactions, the exogenous administration of this molecule is able to increase affiliative social behavior, such as exchanging glances and contact seeking [63,64]. By contrast, the exogenous administration of oxytocin under different conditions, including threat, has been shown to produce an inhibitory effect on dogs’ friendly behavior toward humans [65], and a “not always prosocial” effect for oxytocin as was proposed [13,66,67].

In the early stages of domestication, the establishment of conditions in which some populations of wolves and humans maintained regular physical proximity was essential [13], and humans led a highly dynamic lifestyle. It is likely that they exploited many of the same territories occupied by wolves. Recent studies show that wolf packs tend to regulate their activity within a territory in order to avoid human encroachment [68]. Such conditions may have created chronic stressors for some wolf populations, which started to reduce their land use or adopt behavioral strategies to coexist in human-trafficked areas [13,69]. Over generations, these wolves may have been more inclined to occupy territories in proximity to human settlements, raising offspring that showed further reductions in fear and anxiety through exposure to the anthropogenic environment. It is also possible that the anthropic niche has favored a transition from the “highly selective” sociality observed in wolves to a less structured social system, in which selective sociality (including monogamous reproduction) has become less crucial, perhaps in response to the transition from cooperative hunting to solitary scavenging.

## 3. Wolves vs. Dogs: A Socio-Ecological Clash

Wolves are cursorial predators capable of hunting in groups (pack-hunting specialists) that base their diet mainly on ungulates. Their social organization involves a monogamous pair, accompanied by their offspring, composed of adults, subadults and puppies (nuclear family organization), and sometimes, unrelated individuals join the pack [20]. The size of the pack has a fundamental role in the success rates of hunting and in the defense of the territory, and very often clashes between schools result in lethal aggressions. Overall, it seems that wolves are heavily dependent on their social structure as a cohesive and functional pack, which allows them to get enough food, breed their cubs, and defend the territory [23]. As for dogs, free-ranging dogs show a feeding ecology closely related to the occupied niche in the human environment. Their diet consists mainly of human food waste, and in many populations, hunting (including pack hunting) plays a minor role. They have been considered “optionally social”, since the size of the group depends on the abundance of food. In contrast to wolves, they have a much more flexible pack structure, as members of one pack often join another. Regarding the care of the offspring, dogs differ much from wolves. The cubs of free-ranging dogs are bred mainly by mothers [20], while among wolves, the occurrence of cooperative breeding of cubs by the pack with alloparental care is now well known [70].

In support of a reduction in fear related to the domestication process, wolves appear to be more neophobic toward humans and require intense early socialization to foster “confident” man–wolf interactions [20]. Even when trust relationships are established, they are limited to specific caregivers and not easily generalized to outsiders [71,72].

Socialized adult wolves react with fear to humans, and if only partially socialized, fear can turn into aggression [20]. In addition, wolves who have socialized with humans (both puppies and adults) also show a greater neophobia than dogs toward objects [20,23,73,74,75].

However, it must be said that the populations of wolves which have never been hunted by men, such as the Arctic wolves (*C. lupus arctos*), show minimal reactions of fear toward humans and their artifacts [76]. Once the “initial fear” is overcome, wolves seem to be more exploratory [20,23,73,75], persistent in manipulating objects [77], and more prone to risk [78] in comparison to dogs. These findings are very interesting, as they suggest that the differences between the wolf and the dog in terms of fear reactions may be partly explained by the selection for human fear/shyness in wolves, due to the strong persecution that these animals have suffered over time, and not only due to the selection against fear in dogs [20].

## 4. Genetics of Ancient European Dogs

The origin of dogs can be traced back to the ancient coevolution and mutualism established between Paleolithic humans and wolves. Archaeological and paleontological evidence of ancient canid remains from France, Germany, and Spain indicates that the earliest domestic dogs identified certainty originated at least 15,000 or 13,500 years ago [79,80]. By contrast, some studies have described Paleolithic dogs dating back to 40,000–16,000 years ago from Belgium (Goyet Cave), Russia, Germany and Czech Republic, although the taxonomic attribution of these specimens is still controversial [81], while others have redefined the time interval within which the phenomenon of domestication is likely to have occurred [82,83,84]. The archeozoological analysis of fossils of large canids unearthed from different excavation sites located in Moravia (Eastern Czech Republic) dating back to the Pavlovian (29,000–25,000 years ago) allowed the attribution of these remains to *C. l.* and stated that the Pavlovian hunter-gatherers shared a rather conflicting relationship with these wolves [82]. The clash was probably motivated both by competition for the same trophic resources (overlapping of preys) and survival (overlapping of ranges). Furthermore, the absence of traces of gnawing on the examined fossil bones could be indirect evidence of the absence of domesticated dogs in such human settlements [82]. This scenario is far from being described as an example of ancient coexistence between dogs and humans, and it seems to place the “beginning” of dog domestication much later than 40,000 years ago.

Early genetic research on the origin of *C. f.* focused on the mitochondrial DNA (mtDNA) control region [85,86], as the high mutational rate and maternal inheritance make it an excellent marker for population studies and lineage tracking [87]. The first large-scale study focusing on the mtDNA control region of 140 domestic dogs and wolves was published in 1997 and revealed four different clades [85]. The haplotypes of the dog and wolf differed by a maximum of 12 mutations, while the haplotypes of the dog and coyote/dhole differed by at least of 20 mutations, thus indicating that the wolf is the direct ancestor of the domestic dog [85].

The recent development of sequencing techniques and new protocols for also processing ancient DNA (aDNA) has made it possible to obtain data with a quality never seen before. Several paleogenomics studies have revealed the long-standing history and proximity of the dog to humans during the Paleolithic, supporting predictions based on the Taimyr wolf genome dated to 35,000 years ago [88] and the profound diversification of dogs in the Early Holocene, with the presence at least of five ancestral dog lineages dating back to 11,000 years ago [89]. The study of ancient human genomes has revealed an important transformation in ancestry associated with the expansion of Neolithic farmers from the Near East to Europe [89], and a study of the mtDNA of ancient dogs suggested that such human populations were accompanied by these animals [90]. The western (Anatolian and Levant) and eastern (Zagros Mountains of Iran) human populations of the Fertile Crescent were highly genetically differentiated, and the western groups were the primary source of gene flow into Europe and Africa during the Neolithic period [89]. A source of African canine ancestry from the Levant (7000 years ago) is more suitable than that from Iran (5800 years ago), reflecting human and cattle history, while the expansion of steppe pastoralists associated with the Yamnaya and Corded Ware cultures into the Late Neolithic and Bronze Age Europe profoundly transformed the ancestry of human populations [89]. Comparing genomes from ancient human and canine datasets, Bergström and colleagues have also inferred different dispersal patterns, showing interesting concordances (i.e., ancestors related to the Levant in Africa and early agricultural Europe) or discrepancies (pastoral expansion into the steppes in Eurasia) between the population dynamics of these two species over time and space [89]. They hypothesized that the cline of genomic ancestry observed in ancient European dogs might be due to an admixture between dogs associated with Mesolithic hunter-gatherers and incoming Neolithic farmers. Furthermore, a comparison between modern African and ancient dogs from the Levant and Iran suggested a Near-Eastern origin [89].

Despite the potential correlation between the steppe and the Corder Ware culture [91], most of the later European dogs did not show clear affinities with the Srubnaya culture [89], while modern European dogs clustered with Neolithic European dogs and did not reflect the sustained change in ancestry observed in human populations after pastoral expansion. The relative continuity between Neolithic and present-day individuals suggests that the arrival of steppe pastoralists did not result in large-scale persistent changes in the genetics of European dogs [89], and a single [89,91] or dual [92] origin for domestic dogs is supposed.

In general, ancient dogs are more genetically diverse than modern dogs, and this wide genetic diversity disappeared long before the Victorians started creating new breeds [89]. Despite all European dogs appearing to have descended from one group of ancient European dogs, the great diversity in shape and size among modern dogs indicates a human selection focused on certain genes. The first modern dog genome to be sequenced belonged to a male poodle, with a coverage of 1.5×; however, large regions of the genome were not sequenced, resulting in a fragmentary genome and the impossibility of identifying with certainty the sequences of the canine Y chromosome [93]. The second attempt was made on a female boxer. In this case, the canine genome was sequenced with a coverage of 7.5×, obtaining almost the entire genome [94]. The dog genome consists of 39 pairs of chromosomes, and the size of the euchromatic (genetically active) genome has been estimated to be between 2.31 and 2.47 Gb [93,94], with a low proportion of repeat insertions. The rate of SNP (single-nucleotide polymorphism) in the canine genome between different breeds has been estimated at about one SNP per 900 bp, while between boxers and grey wolves it was one SNP per 580 bp and for coyotes 1/420 bp [94].

### 4.1. Canine Y Chromosome Marker

In population studies, the uniparental genetic marker Y chromosome represents a powerful tool for understanding the historical development of dog breeds [95,96]. It is a male-specific chromosome and contains the MSY gene, whose activity induces maleness, and several genes implicated in spermatogenesis [95,97]. The Y chromosome has been employed in many studies concerning animal populations, i.e., as recently reviewed with regard to horses [98] or cattle [99], evaluating a potential founder effect [100,101] and focusing on microsatellites [102] and SNP [103]. Compared to maternal line data, relatively little is known about the diversity of the Y-chromosome haplogroups found among ancient dogs in Europe. As seen in mtDNA analyses, contemporary sample studies show that the phylogenetic tree constructed on the basis of Y chromosome data is characterized by deep splits between dogs [104].

A study based on 151 dogs revealed the existence of five Y-chromosome haplogroups (ChY Hgs) [105], while a more extensive analysis conducted on hundreds of canine samples showed a high diversity of Y-chromosome haplotypes in Africa, India, Central Asia and South-Western Asia [106]. More recently, the diversity of Y-chromosome haplotypes observed among contemporary and ancient dogs was investigated by remapping 151 markers present in the 170 K Illumina HD Canine SNP Array to the assembly of the Y chromosome published by Li et al. [107,108]. Oetjens and colleagues made a phylogenetic reconstruction of 118 canid Y chromosomes and a common ancestor of HG1-3, HG27, and HG6 that arose long after separation from HG9 [108] (Figure 3).

In line with previous analyses of the canid Y chromosome and mtDNA [104,109], among the deepest branches (belonging to the coyote, red wolf and Great Lakes wolf) of the Y-chromosome phylogeny, coyote (*C. latrans*) diverged from the “dog–gray wolf” clade and shared a clade with the red wolf (*C. rufus*), which is known to be characterized by a genome resulting from frequent crossbreeding events with the coyote [110,111], while a wolf sample from the Great Lakes in Minnesota [112] presented an incredibly divergent Y chromosome with 199 unique derived alleles [108]. Moreover, in order to produce a first representation of the diversity of ChY among ancient dogs, Oetjens and colleagues determined the haplotypes of three European fossil samples previously analyzed [91,92] and confirmed a long-lasting population structure for European dogs, with at least two ChY haplogroups present during the Neolithic [108]. These included an ancient dog from the Newgrange tombs complex (NGD; Ireland) and dating back 4800 years, a sample from the Early Neolithic archaeological site in Herxheim (HXH; Germany) dating back to 7000 years ago, and a sample from the Cherry Tree Cave in Bavaria (CTC; Germany) dating back 4700 years. Both CTC and HXH had derived alleles at all the diagnostic sites of HG8–HG23, although sites specifically diagnostic for the HG23 or HG8 clades were not selectable for these two samples. However, CTC had two of the four selectable derived alleles unique to the Indian wolf line (HG23 haplotype), and differently from the NGD Y chromosome which belonged to the HG1-3 lineage, it had no diagnostic alleles that could match contemporary modern dogs or wolves within this haplogroup [108].

### 4.2. A Focus on the Ancient mtDNA Phylogeny in Three Mediterranean Peninsulas

Most recent studies focused on the female counterpart of mtDNA to inspect the evolutionary processes that affected the maternal lineages, and dogs were conventionally grouped into four main mtDNA haplogroups (Hgs): A, B, C, and D.

Europe is one of the most investigated regions concerning ancient dogs, since it was observed that old Europe (from the Pleistocene to the Holocene) presents four regions of dog populations: Northern Europe, presenting both the A and C haplogroups [90,92]; Central/Western Europe, with high frequencies of the Hgs C and D [88,90,91,92,109,113]; Eastern Europe, characterized by the presence of HgD, as the main clade (over 90%), together with A and C [90,92,109]; and Southern Europe, showing the detection of the A, B, and C lineages [90,92,114], with the first record for high frequencies of HgA in pre-Neolithic Europe [115].

In 2016, Frantz and colleagues pointed out that ancient European dogs belonged to HgC or HgD, with HgC being the most frequently observed in Europe before the Neolithic period (more than 8000 years ago); however, most modern European dogs belong to HgA and HgB [92]. The first mtDNA lineage present in Europe before the Neolithic seems to be HgC, while HgA and HgD are believed to have arrived in Europe during Neolithic and post-Neolithic migrations together with humans [90]. It appears that crossbreeding between the dog and the wolf did not contribute significantly to the gene pool of mitochondrial DNA in the domestic dog [116].

Despite scholars still debating the times and places of dog domestication [90,117], it is well-known that the Mediterranean area has played a key role in re-shaping the genetic variability of dogs after the Last Glacial Maximum (LGM; 24,000–18,000 years ago). In this scenario, the Italian (Apennines refuge), Iberian and Balkan peninsulas represent crucial territories due to their geographic position and the presence of three glacial refuges during the LMG. The Italian Peninsula has been the object of different human population studies [118,119,120,121] (as examples), thus being particularly interesting for the analysis of canine population dynamics and migrations [81] strictly connected to human patterns. Recently, a few samples of ancient Italian canids have been genetically analyzed, and these were mainly wolves. In 2019, Ciucani and colleagues demonstrated that mtDNA HgA spread significantly earlier than previously expected by analyzing a short portion of the hypervariable region 1 (HVR-1) of the mtDNA from 19 skeletal remains of Italian canids dating back to the Late Pleistocene, Bronze Age, and Middle Ages from three different locations in Northern Italy [122].

These findings, combined with those of Pires and colleagues highlighting the presence of late Pleistocene wolves and Mesolithic dogs belonging to HgA in another glacial refuge (Iberian Peninsula) [115], stimulated strong interest in the southern area of the European continent and its role within the canine domestication process [81]. Neolithic Bulgarian dogs from the Balkan Peninsula showed a high percentage of HgA [123], while Koupadi and colleagues analyzed the mtDNA HVR-1 region of 27 fossil canid samples and found seven different haplogroups [81].

Based on the subdivision of the two major and distinct wolfish mitochondrial haplogroups (known as Hg1 and Hg2) proposed by Pilot et al. [124], one sequence was attributed to the wolf haplogroup (Hg2), while the remaining samples were assigned to canine haplogroups A–D (five sequences assigned to HgA, four to HgC, and one to HgD). The analysis of canine maternal lines in ancient Eurasian dogs attested to the presence of HgA in Italy dating back more than 15,000 years and the appearance of the HgD haplogroup about 8000 years ago [81].

Phylogenetic data suggested a close relationship between Italian [81], Iberian [115], and Balkan [123] canids, since in different cases they share the same mitochondrial haplotype, and corroborated the theory that these geographical areas have played a crucial role in the dynamics of canine domestication. By analyzing a pool of 97 ancient dog mtDNA samples of different dates (time interval from Late Pleistocene to Late Antiquity) belonging to the three Mediterranean peninsulas (Iberian [115,125,126], Italian [81,114,122,127] and Balkan [80,123]) (Appendix A), we could analyze the abundance and the distribution of canine mitochondrial haplogroups within the three geographic locations of the fossil recoveries (Figure 4).

The high presence of HgB recorded in dogs and wolves from the Balkans seems to indicate that some hybridization events occurred before the Neolithic period [123], while the increasing HgA frequencies follows a geographic gradient from East to West and is proposed to be a consequence of Neolithic farmer migration from the Middle East toward Europe [90]. Nevertheless, its detection in Iberian Mesolithic samples, presenting mitogenomes different from Middle Eastern dogs, seems to evidence a pre-Neolithic local process of Iberian wolf domestication [115]. Although it is believed that the earliest European dog populations belonged to haplogroup C, the presence of an Italian 24,700-year-old canid belonging to HgA [122] raised the need to further investigate the origin of dog domestication in Europe. The comparative analysis of five Bronze-Age samples revealed a correlation with this Italian canid, as they differ only in one nucleotide position, and confirmed the early presence of HgA in Europe [81].

The analysis of the abundance of canine mitochondrial haplogroups as a function of the temporal dating for each peninsula allowed us to construct a graph to observe any changes in the occurrence of the four mtDNA haplogroups during the last 20,000 years (Figure 5). Unfortunately, however, the same time interval could not be analyzed for all the geographic areas because of a slight divergence between the temporal focuses of the studies here considered.

The graphs display high HgC frequencies for the oldest samples found in the Mediterranean area and the strong presence of HgA also until more recent periods, thus confirming a well-defined European genetic dog structure with a clear separation between eastern (clades B and D) and central–western (clade C) Europe.

Moreover, in order to graphically display and summarize the mitogenetic relationships among the three Mediterranean peninsulas in different time periods, a Principal Component Analysis (PCA) was performed using Excel software implemented by XLSTAT and with the haplogroup frequencies used as input data (Figure 6; Appendix A).

A geographic differentiation is clear in the PCA plot, where the contribution of HgA pushes the Iberian samples from the Late Pleistocene into a separate quarter. Except for the Italian Peninsula, which seems to be near to the others depending on the period, the Balkan Peninsula remains dating back from 8500 years ago to the Roman age are separated by the PC2 due to the high contribution of the B and D haplogroups. These results reflect the geographic position of the Italian Peninsula, which is considered a key crossing point in the Mediterranean landscape, and confirm the role of exchanges that have largely influenced the effective gene flow within the native populations of the Mediterranean Basin [128].

## 5. Conclusions and Perspectives

The mechanisms behind canine domestication represent one of the most difficult challenges in the field of evolutionary biology. It was a “multiphase” process, with a first phase during which different groups of wolves were attracted by the anthropogenic niche and a second phase characterized by the gradual establishment of mutual relationships between wolves and humans. Dog (*C. f.*) domestication was subject to different sources of selective pressure caused by both the environment and humans, and various molecular mechanisms influenced the affiliative behaviors first observed in Belyaev’s foxes. Animal’s cognitive abilities are modelled by the social ecology, and the different behavioral attitudes of dogs and wolves seem to be due to the action of oxytocin and the arginine vasopressin neuropeptides. Furthermore, the diversity between wolfs and dogs in terms of fear reactions may be also explained by a selection for human fear/shyness carried out on wolves due to the strong persecution that these animals have suffered over time.

From a genetic point of view, recent paleogenomics studies allowed the reconstruction of the dispersal patterns of Pleistocene canids and adjusted the focus on the exact location of the main centers of canine speciation. The use of uniparental genetic markers has deepened our knowledge of canine phylogeny, finding out the possible existence of genetic admixture events between dogs and wolves through ChrY analysis and redefining the evolutionary history of the genus Canis based on mtDNA studies. In particular, three Mediterranean peninsulas (Balkan, Iberian and Italian) represent the main geographic area for the study of canine domestication dynamics, thus highlighting the spread of HgA during the Neolithic farmer migration from the Middle East and confirming a well-defined European genetic dog structure with a clear separation between eastern (clades B and D) and central–western (clade C) Europe.

Yet, the biological basis for the fascinating phenomenon of dog domestication needs to be further investigated, both to broaden our view of the evolutionary biological picture and to better understand the extent to which humans are actually affecting the fitness of the global ecosystem.

## Figures and Tables

**Figure 1 genes-14-00992-f001:**
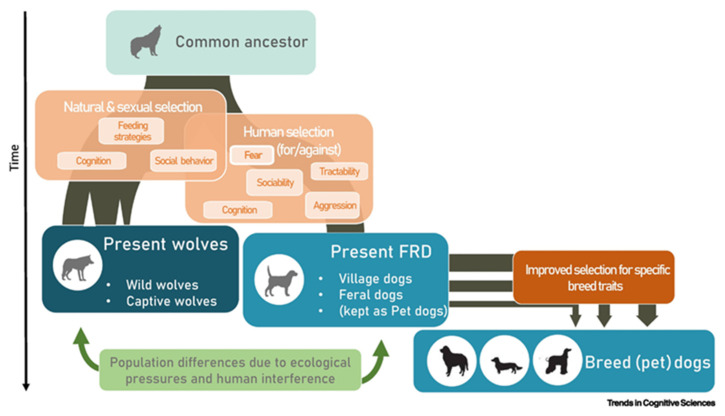
Simplified diagram of the domestication process from [20]. The figure summarizes the wolf domestication process and the respective selective pressures that are likely to have been active during the process. Abbreviation: FRD, free-ranging dogs.

**Figure 2 genes-14-00992-f002:**
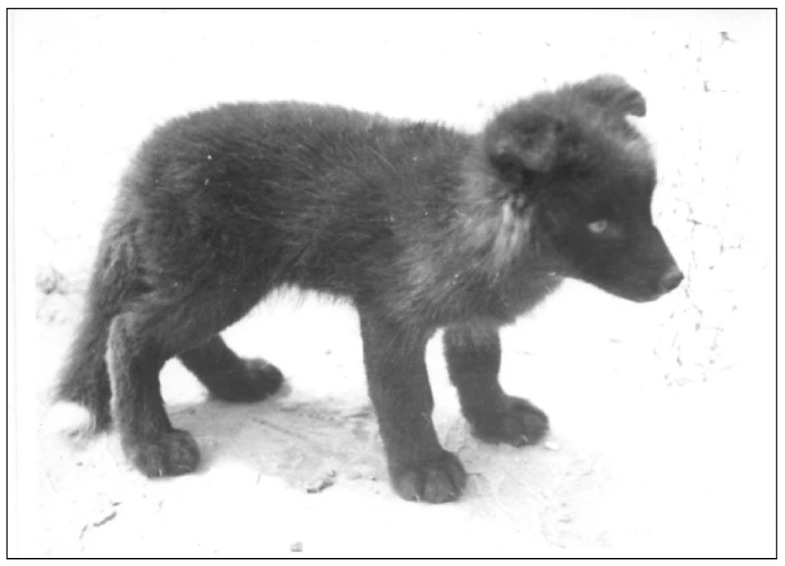
Mechta (Dream), the first of the domesticated foxes to have floppy ears (1969) from [27].

**Figure 3 genes-14-00992-f003:**
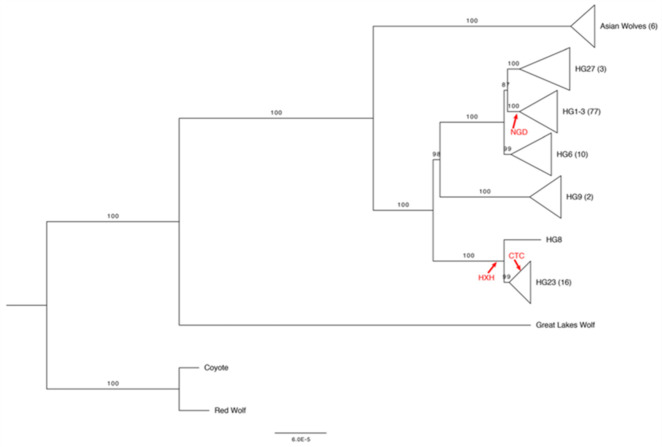
Maximum likelihood phylogeny of 118 canid Y chromosomes from [108]. The clades in the tree have been collapsed by haplogroup assignment. The number of samples within each collapsed node is indicated in parentheses next to the haplogroup assignment. The locations of three ancient samples (NGD: Newgrange grave complex in Ireland; HXH: early Neolithic site of Herxheim in Germany; CTC: Cherry Tree Cave in Bavaria, Germany), as based on the presence of diagnostic mutations, are indicated in red.

**Figure 4 genes-14-00992-f004:**
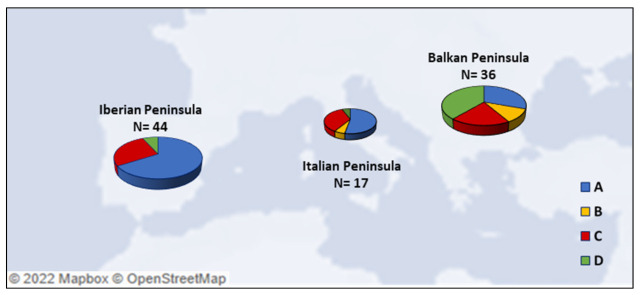
Frequencies of the four main canine mtDNA haplogroups (A, B, C and D) in the three Mediterranean peninsulas. Details are reported in Appendix A.

**Figure 5 genes-14-00992-f005:**
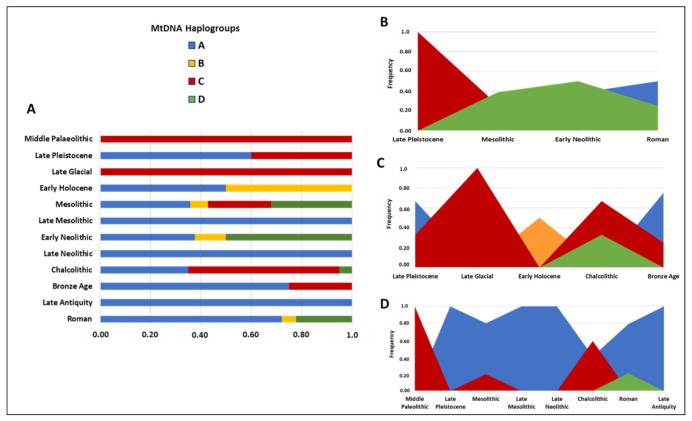
Frequency of the four main mtDNA haplogroups (**A**–**D**) in different time periods (**A**) and their temporal distribution in the Balkan (**B**), Italian (**C**) and Iberian (**D**) peninsulas. The most recent dogs here considered are from Late Antiquity [81]. Details and frequencies are reported in Appendix A.

**Figure 6 genes-14-00992-f006:**
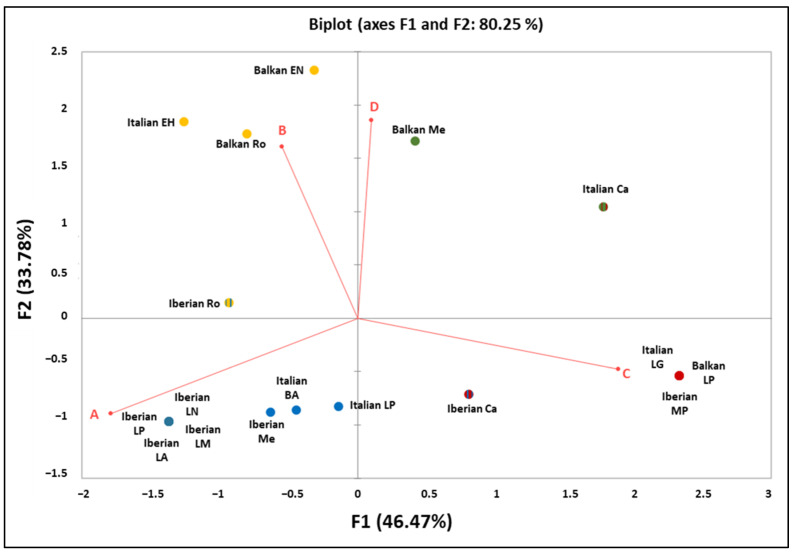
Principal Component Analysis (PCA) plot representing the genetic landscape of the Mediterranean peninsulas, as performed using Excel software implemented by XLSTAT and based on the haplogroup frequencies. Population codes and frequencies details are reported in Appendix A.

## Data Availability

No new data were analyzed in this study. Data sharing is not applicable to this article.

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
