# Peer review of "Being a Dog: A Review of the Domestication Process"

_genes, 2023, doi:10.3390/genes14050992_

Round 1
Reviewer 1 Report
Dear authors,
the article is very well written and prepared in editorial and scientific terms.
On the controversy over the time of domestication of the dog, he suggests including work:
Wilczynski, J.; Haynes, G.; Sobczyk, L.; Svoboda, J.; Roblickova, M.; Wojtal, P. Friend or foe? Large canid remains from Pavlovian sites and their archaeozoological context. J. Anthropol. archaeol. 2020, 59, 101197. [https://doi.org/10.1016/j.jaa.2020.101197].
Alternatively, but only if the authors deem it necessary, several similar works may be included, which are voices in the polemic concerning this work.
The manuscript reads very well, its individual parts are clearly separated and the research problem is presented and discussed in a logical way. It is presented in a way that is easy to understand even by people who are not related to the subject and have no experience in this matter. It should also be emphasized here that the authors approached the analysis of the problem critically and showed the controversies regarding this topic.
In the attached manuscript, I have noted minor stylistic errors, which, however, do not affect my positive reception of the work.

Author Response
Dear authors,
the article is very well written and prepared in editorial and scientific terms. On the controversy over the time of domestication of the dog, he suggests including work:
Wilczynski, J.; Haynes, G.; Sobczyk, L.; Svoboda, J.; Roblickova, M.; Wojtal, P. Friend or foe? Large canid remains from Pavlovian sites and their archaeozoological context. J. Anthropol. archaeol. 2020, 59, 101197. [https://doi.org/10.1016/j.jaa.2020.101197].
Alternatively, but only if the authors deem it necessary, several similar works may be included, which are voices in the polemic concerning this work.
- Authors’ response: we would like to thank the reviewer for this suggestion. We included Wilczynski et al. 2020 and other references in the manuscript.
The manuscript reads very well, its individual parts are clearly separated and the research problem is presented and discussed in a logical way. It is presented in a way that is easy to understand even by people who are not related to the subject and have no experience in this matter. It should also be emphasized here that the authors approached the analysis of the problem critically and showed the controversies regarding this topic.
- Authors’ response: we would like to sincerely thank the reviewer for his positive comments.
In the attached manuscript, I have noted minor stylistic errors, which, however, do not affect my positive reception of the work.
- Authors’ response: we have modified the manuscript by following the Reviewer’s suggestions about some minor stylistic errors.
Reviewer 2 Report
The article intends to review literature relating to evolution of the domestic dog.
" Here, we provide a review of the process of dog (Canis familiaris) domestication, which was subjected to different sources of selective pressure caused by both the environment and humans, highlighting the ecological differences between dogs and wolves, and analyzing the molecular mechanisms which influenced the affiliative behaviors firstly observed in Belyaev's foxes."
To begin, this purpose does not make sense and has several grammatical issues. The focus on the Mediterranean peninsulas seems to be poorly connected.
Unfortunately, in its current form the review is poorly structured and lacks critical insight. It contains a moderate number of grammatical errors throughout. Figures are not original but are properly attributed. It is however unclear what the images are adding to the comprehension of the topic. There is no thorough questioning or alternate views on hypotheses presented as accepted (such as domestication syndrome). For example an important refutation of the domestication syndrome hypothesis by Lord et al (2020) is missing and the review appears to lean heavily on the work of Herbeck et al (2022).
The reference genome sequencing section appears to be improperly placed and there seems to be a heavy reliance on mitochondrial sequence later. In general, the review spends much more time discussing behavioral and social changes from wolf to dog than genetic changes.
This is a complex topic and one that in the dog genomics world has inspired robust debate. This is acknowledged in another publication including the same team in the author list but not really discussed here.
Perhaps this article would read better if the reader was immediately informed that the intention was to focus on the evolution of dogs of the Mediterranean basin. As it is, it feels only half done.
As a review paper, I would not have expected data to be included. The methods for the analysis depicted in figures 4,5 and 6 do not seem readily available.
Author Response
The article intends to review literature relating to evolution of the domestic dog.
" Here, we provide a review of the process of dog (Canis familiaris) domestication, which was subjected to different sources of selective pressure caused by both the environment and humans, highlighting the ecological differences between dogs and wolves, and analyzing the molecular mechanisms which influenced the affiliative behaviors firstly observed in Belyaev's foxes."
To begin, this purpose does not make sense and has several grammatical issues. The focus on the Mediterranean peninsulas seems to be poorly connected.
- Authors’ response: we would like to thank the reviewer for his comments. We modified the abstract as follows “The process of canine domestication represents certainly one of the most interesting questions that evolutionary biology aims to address. A "multiphase" view of this process is now accepted, with a first step during which different groups of wolves were attracted by the anthropogenic niche, and a second phase characterized by a gradual establishment of mutual relationships between wolves and humans. Here we provide a review of dog (Canis familiaris) domestication, highlighting the ecological differences between dogs and wolves, analyzing the molecular mechanisms which seem to have influenced the affiliative behaviours firstly observed in Belyaev's foxes, and describing the genetics of ancient European dogs. Then we focused on three Mediterranean peninsulas (Balkan, Iberian and Italian), which represent the main geographic area for studying canine domestication dynamics, since it has shaped the current genetic variability of dog populations, and where a well-defined European genetic structure was pinpointed through the analysis of unip-arental genetic markers and their phylogeny.”. We decided to focus on the Mediterranean area, which represents a key region for the genetic shaping of European dogs.
Unfortunately, in its current form the review is poorly structured and lacks critical insight. It contains a moderate number of grammatical errors throughout. Figures are not original but are properly attributed. It is however unclear what the images are adding to the comprehension of the topic. There is no thorough questioning or alternate views on hypotheses presented as accepted (such as domestication syndrome). For example an important refutation of the domestication syndrome hypothesis by Lord et al (2020) is missing and the review appears to lean heavily on the work of Herbeck et al (2022).
- Authors’ response: we would like to thank the reviewer for his comments. We have now improved the manuscript aiming to approach the analysis of the problem critically and show the controversies regarding this topic. For this reason, we have modified the manuscript by adding also other references and including the refutation of the domestication syndrome hypothesis by Lord et al. 2020.
The reference genome sequencing section appears to be improperly placed and there seems to be a heavy reliance on mitochondrial sequence later. In general, the review spends much more time discussing behavioral and social changes from wolf to dog than genetic changes. This is a complex topic and one that in the dog genomics world has inspired robust debate. This is acknowledged in another publication including the same team in the author list but not really discussed here.
- Authors’ response: we would like to thank the reviewer for his comments. We have moved the reference genome sequencing section at the end of the paragraph. Moreover, we decided to discuss also behavioral and socio-ecological changes from wolf to dog because they have contributed to shape genetic variability and features.
Perhaps this article would read better if the reader was immediately informed that the intention was to focus on the evolution of dogs of the Mediterranean basin. As it is, it feels only half done. As a review paper, I would not have expected data to be included. The methods for the analysis depicted in figures 4,5 and 6 do not seem readily available.
- Authors’ response: we would like to thank the reviewer for his comments. We have added methods for the analysis depicted in Figure 6 by modifying the legend as follows “Figure 6. Principal Component Analysis (PCA) plot representing the genetic landscape of Mediterranean peninsulas, performed using the Excel software implemented by XLSTAT and based on haplogroup frequencies. Population codes and frequencies details are reported in Table S3”. Further details for each figure are reported in three supplementary tables (Tables S1, S2 and S3).
Reviewer 3 Report
The paper “Being a dog: a review of the domestication process” represents a very interesting, comprehensive and updated analysis of the current state of knowledge about dog domestication. This paper is very well written, is easy to follow and the analysis back up their results.
I have some minor points that I suggest the authors to take into account to increase the overall clarity of the paper. Please find my suggestions below.
- Please reconsider the sentences about Belyaev's experiment and domestication syndrome in the light of the criticisms highlighted by Lord et al. 2020 ( The history of farm foxes undermines the animal domestication syndrome).
- L. 26-28 pag. 1: I suggest to use a different definition of domestication such the one proposed by Clutton-Brock (A Natural History of Domesticated Mammals)
- L. 248” wolf “ without capital letter and also please add the correct reference (Skoglund et al. 2015) instead of [72]
- L. 344 it is not clear the association between the studies cited by the authors and the dog domestication topic, especially regarding the chronology of some cited papers. Please revise and clarify this sentence.
- L.355-358 the sentence is not clear, please rephrase it. If I am not wrong Pires doesn’t state that “hg A arrived in Europe only during the Bronze Age, about 3,000 years ago”
- L.383 The absence of HgB in Iberia could not directly correlate to hybridization events. Rephrase the sentence or justify this statement
- L 385-387 it is not clear how the observed gradient of hg A could be correlated to Neolithic migration, please add an explanation of that or rephrase the sentence
Author Response
The paper “Being a dog: a review of the domestication process” represents a very interesting, comprehensive and updated analysis of the current state of knowledge about dog domestication. This paper is very well written, is easy to follow and the analysis back up their results.
- Authors’ response: we would like to sincerely thank the reviewer for his positive comments.
I have some minor points that I suggest the authors to take into account to increase the overall clarity of the paper. Please find my suggestions below.
- Please reconsider the sentences about Belyaev's experiment and domestication syndrome in the light of the criticisms highlighted by Lord et al. 2020 (The history of farm foxes undermines the animal domestication syndrome).
- Authors’ response: we would like to thank the reviewer for his comments. We have modified the manuscript by discussing also other references and including the refutation of the domestication syndrome hypothesis by Lord et al. 2020.
- L. 26-28 pag. 1: I suggest to use a different definition of domestication such the one proposed by Clutton-Brock (A Natural History of Domesticated Mammals)
- Authors’ response: we have modified the definition of domestication as follows: “It arises from a mutualism between two species, where the domesticator creates an environment and actively manages both survival and reproduction of another species (the domesticated), which provides the former with resources and/or services [Purugganan et al. 2022].”
- L. 248” wolf “ without capital letter and also please add the correct reference (Skoglund et al. 2015) instead of [72]
- Authors’ response: we changed "Wolf" to "wolf" and added the correct reference as recommended.
- L. 344 it is not clear the association between the studies cited by the authors and the dog domestication topic, especially regarding the chronology of some cited papers. Please revise and clarify this sentence.
- Authors’ response: we have revised the sentence as follows “The Italian peninsula was object of different human population studies [Fiorito et al. 2016; Raveane et al. 2019; Modi et al. 2020; Aneli et al. 2021] (as examples), thus resulting particularly interesting for the analysis of canine population dynamics and migrations [72], strictly connected to human patterns.”.
- L.355-358 the sentence is not clear, please rephrase it. If I am not wrong Pires doesn’t state that “hg A arrived in Europe only during the Bronze Age, about 3,000 years ago”
- Authors’ response: thank you for the suggestion. This was a clear mistake. We have modified the sentence as follows “Neolithic Bulgarian dogs from the Balkan Peninsula showed with a high percentage of HgA [Yankova et al. 2019], while Koupadi and colleagues analyzed the mtDNA HVR-1 region of 27 fossil canid samples and found seven different haplogroups [Koupadi et al. 2020].”.
- L.383 The absence of HgB in Iberia could not directly correlate to hybridization events. Rephrase the sentence or justify this statement
- Authors’ response: we have modified the sentence as follows: “The high presence of HgB recorded in dogs and wolves from Balkans seem to indicate some hybridization events occurred before the Neolithic period [Yankova et al. 2019], while the increasing of HgA frequencies follows a geographic gradient from East to West and is proposed to be a consequence of Neolithic farmer migration from the Middle East toward Europe [Ollivier et al. 2018]. Nevertheless, its detection in Iberian Mesolithic samples, presenting mitogenomes different from Middle Eastern dogs, seems to testify for a pre-Neolithic local process of Iberian wolf domestication [Pires et al. 2019]”.
- L 385-387 it is not clear how the observed gradient of hg A could be correlated to Neolithic migration, please add an explanation of that or rephrase the sentence
Authors’ response: we have modified the sentence as follows: “The high presence of HgB recorded in dogs and wolves from Balkans seem to indicate some hybridization events occurred before the Neolithic period [Yankova et al. 2019], while the increasing of HgA frequencies follows a geographic gradient from East to West and is proposed to be a consequence of Neolithic farmer migration from the Middle East toward Europe [Ollivier et al. 2018]. Nevertheless, its detection in Iberian Mesolithic samples, presenting mitogenomes different from Middle Eastern dogs, seems to testify for a pre-Neolithic local process of Iberian wolf domestication [Pires et al. 2019]”.